# Investigation of a Promoted You Only Look Once Algorithm and Its Application in Traffic Flow Monitoring

**Chang-Yu Cao [1], Jia-Chun Zheng [2],\*, Yi-Qi Huang [1], Jing Liu [2] and Cheng-Fu Yang [3],\***

1.  Navigation Institute, Jimei University, Xiamen 361021, China
2.  School of Information Engineering, Jimei University, Xiamen 361021, China
3.  Department of Chemical and Materials Engineering, National University of Kaohsiung, Kaohsiung 811, Taiwan
\*   Correspondence: jchzheng@jmu.edu.cn (J.-C.Z.); cfyang@nuk.edu.tw (C.-F.Y.)



**Featured Application: In this paper, deep convolutional neural networks are applied to traffic flow monitoring and statistics from roads in different scenarios and weather conditions. This study can be used to inform the development of intelligent transportation systems. While using data from traffic flow monitoring and statistical analysis, transportation departments can decide to expand roads or restrict vehicle access to prevent traffic congestion and traffic accidents.**

**Abstract:** We propose a high-performance algorithm while using a promoted and modified form of the You Only Look Once (YOLO) model, which is based on the TensorFlow framework, to enhance the real-time monitoring of traffic-flow problems by an intelligent transportation system. Real-time detection and traffic-flow statistics were realized by adjusting the network structure, optimizing the loss function, and introducing weight regularization. This model, which we call YOLO-UA, was initialized based on the weight of a YOLO model pre-trained while using the VOC2007 data set. The UA-CAR data set with complex weather conditions was used for training, and better model parameters were selected through tests and subsequent adjustments. The experimental results showed that, for different weather scenarios, the accuracy of the YOLO-UA was ~22% greater than that of the YOLO model before optimization, and the recall rate increased by about 21%. On both cloudy and sunny days, the accuracy, precision, and recall rate of the YOLO-UA model were more than 94% above the floating rate, which suggested that the precision and recall rate achieved a good balance. When used for video testing, the YOLO-UA model yielded traffic statistics with an accuracy of up to 100%; the time to count the vehicles in each frame was less than 30 ms and it was highly robust in response to changes in scenario and weather.

**Keywords:** intelligent transportation; traffic flow; promoted You Only Look Once (YOLO) model; loss function

## 1. Introduction

The continually rising number of vehicles on roads has increased the incidence of traffic accidents and traffic congestion, with serious negative effects on people's transportation experiences and overall lives. In light of these changes, an Intelligent Transportation System (ITS) has been rapidly developed and it has attracted significant attention from traffic management departments [1–4]. These departments make decisions largely on the basis of data on road traffic flow. According to the data of traffic flows, the traffic department can know the flowing information on the roads. By determining whether

particular roads have high flow and they are congested, they can provide better real-time guides for drivers. Hence, traffic flow statistics have very important practical applications.

When video surveillance systems are employed on urban roads, the statistics that are obtained from the resulting video images are simple and convenient to use, and the detection and statistic amounts of data are very large, which will not affect the normal travels of vehicles. In probability theory and statistics, a Gaussian model is a stochastic process, because it can collect random variables that are indexed by time or space. A Gaussian model is a simple, easy method for generating background images. When background images are processed while using a Gaussian background updating method, foreground targets can be imaged using a background difference method, with shadow elimination and morphological processing to reduce the illumination intensity and noise [5,6]. This has yielded more accurate traffic statistics, but the statistical accuracy is still not ideal when the traffic flow is large. Previously, Li et al. introduced the Vi Be algorithm, which uses the first frame of a video to initialize the model, sets the foreground threshold and background candidate conditions, updates the model, and extracts the background from the second frame of the video [7]. However, this method is too reliant on relative experiences, and it needs to scan and judge the video image several times when it is used to identify a pixel.

Tan and Dong trained a vehicle classifier to perform vehicle detection on a set area of a video [8]. This method is simple and easy to implement, but only few factors are considerably used in this study, and the classifier is not in general. The studies in ref. [9,10] describe the virtual coil method, which manually sets candidate areas in a video, detects and counts traffic flow by changing the coil state, simplifies calculations, and protects road surfaces, but it is not very effective at detecting multi-lane traffic flows. Other studies examine vehicle-flow statistics that are based on a target tracking method [11–13]. Vehicles detected between adjacent frames in a video are matched and tracked by their specific characteristics to obtain each vehicle's motion trajectory. The images that were obtained from the trajectories are used to detect and calculate traffic flow, but most of the data are noise caused by non-motorized movements and other factors. These results indicate problems with current methods for gathering traffic-flow statistics.

Today, with the aid of artificial intelligence, deep learning is used for target detection, semantic segmentation, image classification, and other identification tasks in various scenarios. The region-based method is the most common for target detection, while using the R-CNN, SPP-net, Fast R-CNN, and Faster R-CNN algorithms [14–21]. R-CNN uses a selective search to extract regions from an image, which is efficient, but training is cumbersome, and its test speed is slow [14,22]. SPP-net makes the network input unrestricted, but the training features are stored on disk, which limits the detection speed [15]. Fast R-CNN combines the R-CNN and SPP-net concepts and it can improve detection speed and accuracy, but the speed is still relatively slow [15,16]. Faster R-CNN uses an RPN network instead of selective search algorithms, which greatly shortens the time to extract regions from images, but only achieves seven frames per second for video detection, falling short of real-time detection requirements [20–22]. In order to improve the detection speed and accuracy, based on the DenseNet model [23], the researchers proposed a lightweight PeleeNet model for mobile devices, which had the highest target classification accuracy [24]. Their further study had combined the PeleeNet and optimized SSD (Single Shot MultiBoxDetector) to develop a real-time target detection system for mobile devices, which had low computational cost and reliable targets' detection performance [24,25]. The ideas have provided very good help and inspiration to subsequent researchers. Next, this paper proposed a method to detect the pedestrian features, which combined a histogram of the oriented gradient (HOG) and discrete wavelet transform (DWT). The method uses the motion amplitude to set the interest region to improve the detection speed. HOG and DWT systems are used to detect pedestrian multi-features, and ROI is then classified by SVM with multi-feature mechanisms, and the detection speed can be improved [21,26]. Subsequently, a network model is proposed for the target detections of lightweight RFBNet, which combines the enhancements of speed and accuracy [27]. The model proposes adding dilated convolution to form a Receptive Field Block (RFB) module, which

is based on the inception structure to increase the receptive field, and introduces RFB into the SSD network to enhance the extraction capability of the network receptive field [25,27]. At the same time, it redefines target detection as a large-scale, but not a sparse distribution problem of boundary box probability, and that proposes the directed evaluation mechanism of the sparse sampling distribution. That can be applied to the end-to-end detection model, which improves the detection performance of the model, but the detection accuracy is not satisfactory [21,28]. In view of this real-time monitoring problem, we need an easily optimized model, with an algorithm that is fast and uses relatively few calculations. In this paper, we present YOLO-UA, which is a regression-based, high-performance algorithm for real-time detection and statistics gathering from vehicle flows that use the You Only Look Once (YOLO) algorithm as the base [29–31]. We employed the Intersection Over Union (IOU) metric and the Generalized Intersection Over Union (GIOU) metric to optimize the loss function directly, and we show that the YOLO model can be modified and promoted to enhance traffic flow monitoring [31,32]. The method of fine-tuning the model structure and the GIOU optimization loss function is proposed to enhance the accuracy of the target positioning in order to solve the problem of poor positioning of the YOLO model and low accuracy of vehicle statistics. After optimization, it can be more reliably applied to video vehicle statistics in real-time and actual scenes. We achieve superior traffic flow detection through optimizing the model and algorithm.

## 2. YOLO Algorithm Statistics and Optimization

YOLO is a single convolutional network that can simultaneously predict multiple bounding boxes and class probabilities for objects in a clear and simple way. YOLO trains on full images and directly optimizes the detection performance. As a one-stage algorithm that combines target location and target recognition into one end-to-end detection process, YOLO can take both speed and accuracy into account [30,31]. The YOLO network is trained by data sets, and the model tests traffic flow statistics. The statistical results reflect the effect of traffic flow monitoring. Figure 1 shows the monitoring architecture.

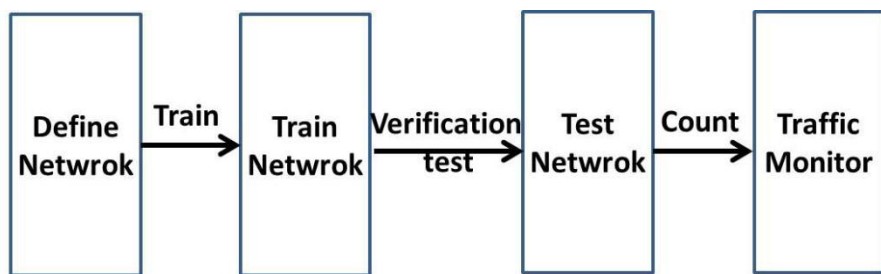

**Figure 1.** You Only Look Once (YOLO) monitoring architecture.

### 2.1. YOLO Network Introduction

YOLO consists of multiple convolutional layers, pooled layers, and fully connected layers. The convolutional layer uses $1 \times 1$ and $3 \times 3$ alternate convolution kernels to extract more abstract features. First, the network is initialized while using the pre-training model of the ImageNet classified data set. Second, because YOLO only supports single-size image input, the original image size is adjusted to $448 \times 448$, and the pixel value is normalized [−1, 1]. Third, the multi-level cascade convolutional layers extract the target feature, use the pooling layer to reduce the dimensions of the target feature, and then send the feature to the fully connected layer. Finally, the fully connected layer predicts the object class and coordinates. The number of neurons in the last layer of the fully connected layer is $7 \times 7 \times 30$ [29–31]. Figure 2 shows the network structure.

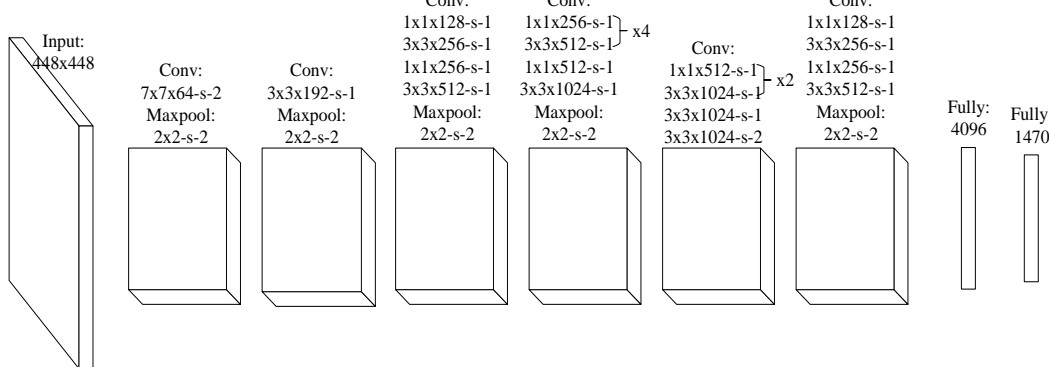

**Figure 2.** YOLO network structure diagram.

## 2.2. Monitoring Principle

The YOLO algorithm is only used for target detection; it does not have a statistical function. To obtain traffic flow statistics, we first need to achieve accuracy while using YOLO to detect the traffic flow, and then we can obtain useful statistics. The monitoring principles are as follows:

(1) YOLO divides each input image into $S \times S$ ($S = 7$) grids, each of which is responsible for predicting the targets that fall into the center of the grid, as shown in Figure 3 [29,31].

(2) The convolutional network extracts features. Each grid aims to predict $B$ ($B = 2$) bounding boxes and $C$ shared class probabilities for a detected object, and each bounding box contains five parameters (*conf*, $t_x$, $t_y$, $w$, $h$), where: *conf* is the bounding box containing the target's confidence value, which is equal to $P(object)^* \text{IOU}_{pre}^{tru}$ if the target $P(object)$ is 1 in the grid, but otherwise is equal to 0; $\text{IOU}_{pre}^{tru}$ is the intersection over union of the prediction box and the real box; ($t_x$, $t_y$) are the offsets from the center of the prediction box relative to the upper left corner of the cell; and, $w$ and $h$ are the ratios of the prediction box to the original image. The final output of the network is a form of the vector of $S \times S \times (5B + C)$ [30,31,33].

(3) After the $S \times S \times B$ bounding boxes are predicted, the confidence values are multiplied by the respective predicted class probability to obtain the class confidence [31,33]. The classification threshold (CT) of the class confidence is set to filter the bounding boxes and the classes whose class confidence is less than that of CT.

(4) Next, we set the the Non-maximum suppression algorithm threshold (NT) and use it to filter the redundant frame for the reserved bounding box [15,16,20,21,31,33].

(5) The final vehicle test results are counted to achieve vehicle-flow monitoring.

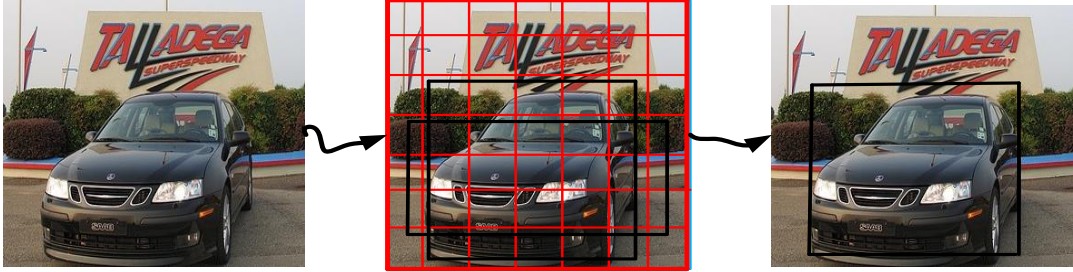

**Figure 3.** Detection principle.

## 2.3. Algorithm Optimization

YOLO can predict only two bounding boxes per grid, which is lower than the number of Faster R-CNN anchor boxes [21]. In a small number of cases, it can roughly detect the location of the target

in the image or video, which causes problems, such as repetition of traffic flows or leakage statistics, which affects the statistical accuracy. Hence, it is very important to use another algorithm to improve the detection accuracy. Previously, Fan et al. proposed using a deep residual network to enhance the automatic learning of hierarchical aesthetics representation; they found that network performance increased with the number of layers [34,35]. For this reason, we tried adding a layer of fully connected layers that are based on the YOLO network and used the dropout function to prevent overfitting.

The regression bounding box is the most important part of computer vision; target detection and tracking depend on having an exact regression bounding box. The traditional methods used to enhance network performance for detection and tracking are deeper backbone networks or better methods to extract local features [34,36,37]. However, both of the methods ignore the optimization of loss functions. IOU is an evaluation metric that can be used to measure the accuracy of an object detector in a particular dataset; it can also be used as an overlapping region metric that is insensitive to scale changes and non-negative [21,31]. If IOU can be used to optimize the loss function directly, it is easy to solve the problems as when that the two boxes were non-intersected and IOU is set to equal to 0, which is equivalent to the loss function without optimization. In addition, IOU cannot address the problem of two-frame overlapping alignment, as shown in Figure 4. The black solid wire frame is the target frame, and the black area is where intersection occurs. In the three cases, the IOUs are the same, but the overlapping modes are different; the effect on the left is the best, and the effect on the right is the worst.

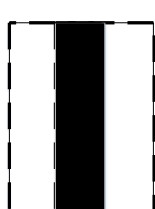 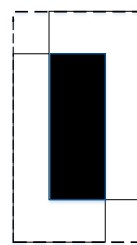 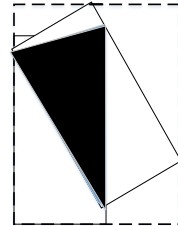

**Figure 4.** Overlap in different ways.

Although IOU is the most popular evaluation metric used in detection benchmarks for moving objects, it has an apparent difference between common optimization used distance losses to regress the parameters of a bounding frame and maximizing this metric value. We try to optimize the loss function of IOU and solve location problems in view of these shortcomings. GIOU metric parameters are proposed for solving location problems, i.e., and GIOU is defined as that IOU subtracts the proportion of non-two-frame area to the minimum closure area [32]. In this case, the minimum closure area is surrounded by the minimum coordinates of the upper left corner of the two frames and the maximum coordinates of the lower right corner of the two frames, as shown by the black dotted line in Figure 4. The range of values is set between –1 and 1. Let the predicted coordinates be $\left(x^p_{\min}, y^p_{\min}, x^p_{\max}, y^p_{\max}\right)$ and the real coordinates be $\left(x^t_{\min}, y^t_{\min}, x^t_{\max}, y^t_{\max}\right)$. Subsequently, the intersection area of the two frames can be calculated while using Equations (1)–(3).

$$x^i_1 = \max(x^p_{\min}, x^t_{\min}), y^i_1 = \max(y^p_{\min}, y^t_{\min}) \tag{1}$$

$$x^i_2 = \min(x^p_{\max}, x^t_{\max}), y^i_2 = \min(y^p_{\max}, y^t_{\max}) \tag{2}$$

$$S_i = (x^i_2 - x^i_1) \times (y^i_2 - y^i_1) \tag{3}$$

Next, Equation (2) can be used to calculate the areas of the prediction frame and the real frame, and the IOU values can be calculated while using Equations (4)–(6).

$$S_p = (x^p_{\max} - x^p_{\min}) \times (y^p_{\max} - y^p_{\min}) \tag{4}$$

$$S_t = (x^t_{max} - x^t_{min}) \times (y^t_{max} - y^t_{min}) \tag{5}$$

$$IOU = S_i/(S_p + S_t - S_i) \tag{6}$$

When we used the IOU value to subtract the weight of the non-two-box merged region in the minimum closure area, the GIOU value could be obtained while using Equations (7)–(10):

$$x^c_1 = \min(x^p_{min}, x^t_{min}), y^c_1 = \min(y^p_{min} - y^t_{min}) \tag{7}$$

$$x^c_2 = \max(x^p_{max}, x^t_{max}), y^c_2 = \max(y^p_{max} - y^t_{max}) \tag{8}$$

$$S_c = (x^c_2 - x^c_1) \times (y^c_2 - y^c_1) \tag{9}$$

$$GIOU = IOU - \frac{S_c - (S_p + S_t - S_i)}{S_c} \tag{10}$$

GIOU is similar to IOU, but it can be regarded as a parameter of overlap measurement, which reflects the overlap relationship between the prediction box and real box, and its value is always smaller than that of IOU [21,31,32]. If GIOU is greater than 0, this means that the two boxes intersect; when it is 1, the two boxes overlap completely. Therefore, when GIOU is equal to 0, the two frames are recognized as infinitely close. If GIOU is less than 0, the two frames do not intersect, and when it is −1, an infinite distance separates the two frames [32]. GIOU focuses not only on overlapping areas, but also on non-overlapping areas, which can reflect how the two boxes actually overlap with each other. In fact, the GIOU value increases as the overlapping area of the two boxes increases. If (1–GIOU) is added to the network loss function, the value of the loss function decreases with the number of iterations, and the degree of overlap becomes larger and larger. Consequently, this definition can effectively solve the problem of inaccurate positioning that results from the YOLO algorithm. L2 regularization is used to enhance the generalization of the network to prevent the over-fitting problem, as Equation (11) shows.

$$LOSS_{total} = loss + \beta \sum_i w_i^2 \tag{11}$$

In Formula (11), $\beta$ is a hyper parameter, and the larger its value, the more important the regularization; $\sum$ is the sum of the squares of the weights. At this point, the network weight is updated, as follows:

$$\begin{aligned} w_{new} &= w - \mu \cdot \frac{\partial LOSS_{total}}{\partial w} \\ &= w - \mu \cdot \frac{\partial loss}{\partial w} - \mu \cdot \frac{\partial(\beta w^2)}{\partial w} = (1 - 2\mu\beta) \cdot w - \mu \frac{\partial loss}{\partial w} \end{aligned} \tag{12}$$

In Equation (12), $\mu$ is the learning rate of the network and regularization can be achieved by multiplying (1–2 $\mu\beta$) as the ownership is re-updated. We designate this optimized algorithm model YOLO-UA.

## 3. Dataset and Network Training

### 3.1. Dataset Production

The new large-scale DETection and tRACking (UA-DETRAC) at State University of New York's University at Albany is a challenging real-world multi-object detection and multi-object tracking benchmark suite [38]. We used the UA-DETRAC data sets to detect traffic on a road overpass. The height and angle of each shot were different to improve the model's adaptability. To study the monitoring of traffic flows in actual scenes, we collected 26,820 images from UA-DETRAC data sets to make UA-CAR data sets, which were then divided into four subsets: training verification set, training set, verification set, and test set. The training verification set accounted for 80% of the UA-CAR data set, the training set accounted for 80% of the training validation set, and they were labeled with LabelImg to generate xml files that corresponded to the images' names. The training verification set initialized a

three-dimensional label in the form of $7 \times 7 \times 25$ with 0 for each image, with 0 list confidence, 1–4 list central coordinates $(x_c, y_c. \ w, h)$, and 5–24 list object class serial numbers. When the xml files were parsed, we extracted all of the target categories in the files and their upper left and lower right corner coordinate values $(x_{\min}, y_{\min}, x_{\max}, y_{\max})$; these data were then multiplied while using ratio values according to the $448 \times 448$ image scaling factor to obtain $(x1_{\min}, y1_{\min}, x1_{\max}, y1_{\max})$. Subsequently, using Equations (13) and (14), the coordinates were converted into the central point coordinate form, and Equation (15) was used to calculate which grid the target center fell into. The grid confidence was set to 1 in the image label, the coordinates of the central point were set to the calculation result of Formulas (13)–(15), and the corresponding target category index was set to 1.

$$x_c = (x1_{\min} + x2_{\max})/2, w = x2_{\max} - x1_{\min} \tag{13}$$

$$y_c = (y1_{\min} + y2_{\max})/2, h = y2_{\max} - y1_{\min} \tag{14}$$

$$x\_ind = [7x_c/448], y\_ind = [7y_c/448] \tag{15}$$

Next, we used horizontal flips to enhance the achievement data and created a dictionary for each image to store the images' paths and labels. Finally, we added all of the dictionaries to the lists and save the lists as pkl files.

*3.2. Network Training*

The YOLO-UA model was initialized while using YOLO model weights by employing the VOC2007 dataset pre-training; the images' paths and labels were extracted from pkl files, and the pixel values of the images were normalized $[-1, 1]$. Iterative trainings were performed 15,000 times while using the stochastic gradient descent algorithm and the initial learning rates were 0.0001. The YOLO-UA loss function consisted of four different parts: coordinate loss, confidence loss, class loss, and GIOU loss (YOLO consisted of only the first three parts), as shown in Equation (16).

$$
\begin{aligned}
loss &= \sum_{i=0}^{48} \sum_{j=0}^{1} ll_{ij}^{obj}(1 - \mathrm{GIOU}) + \gamma_c \sum_{i=0}^{48} ll_i^{obj} \sum_{c \in classes} (p_i(c) - p_i^t(c))^2 \\
&+ \gamma_{coord} \sum_{i=0}^{48} \sum_{j=0}^{1} ll_{ij}^{obj}(x_i - x_i^t)^2 + (y_i - y_i^t)^2 + \gamma_{coord} \sum_{i=0}^{48} \sum_{j=0}^{1} ll_{ij}^{obj}\left(\sqrt{w_i} - \sqrt{w_i^t}\right)^2 + \left(\sqrt{h_i} - \sqrt{h_i^t}\right)^2 \\
&+ \gamma_o \sum_{i=0}^{48} \sum_{j=0}^{1} ll_{ij}^{obj}(C_i - C_i^t)^2 + \gamma_{noobj} \sum_{i=0}^{48} \sum_{j=0}^{1} ll_{ij}^{noobj}(C_i - C_i^t)^2
\end{aligned} \tag{16}
$$

In formula (16), $\gamma_{cood} = 5.0, \gamma_o = 5.0, \gamma_c = 2.0, \gamma_{noobj} = 1.0$, and $(x, y, w, h)$ is the center point coordinate of the offset from the upper left corner of the grid, because a small target is more sensitive to changes in positions. Therefore, the square root of the width and height of the center coordinate was used to calculate the coordinate loss. $(x_i, y_i, w_i, h_i, C_i, P_i(c))$ were the predicted values, $\left(x_i^t, y_i^t, w_i^t, h_i^t, P_i^t(c)\right)$ were the true values, and $ll_{ij}^{obj}$ was the $i$ bounding box of the $j$ grid prediction, which contained the targets [30,31].

## 4. Experiment and Analysis of Results

*4.1. Experimental Platform*

All of the experiments in this paper ran the YOLO algorithm in a Ubuntu 16.04 system, TesnsorFlow 1.8 framework. The program was written in Python 3.5, the processor was an Inter Core i7-7700 CPU, while using a GeForce GTX1080 graphics card to accelerate the training process.

*4.2. Analysis of Experimental Results*

We used the UA-CAR data sets to perform the testing experiments, analyze the experimental data, and compare the experimental results to verify the effectiveness of the optimization. In the process of traffic flow monitoring, there were missed detections and false detections, so we used the precision and recall rate evaluation model. The precision rate was defined as the number of correctly detected statistical vehicles divided by the total number of detected statistical vehicles. The recall rate was defined as the correct number of statistical vehicles divided by the number of vehicles in the data set. The increase in the statistics of correct vehicle detection could improve the recall rate, but, as vehicle misdetection rate increased, the detection's precision would be decreased. On the contrary, the reduction in the number of false vehicle detection was conducive to improving the detection's precision. However, the algorithm could only reduce the number of false vehicle detection when strict judgment conditions were set, but at the same time, it also led to a reduction in the correct statistics of the vehicle, resulting in a lower recall rate. For that, the precision and recall rates showed opposite trends.

*4.3. General Model Experiments*

For calculating the accuracies of the YOLO and YOLO-UA models, CT = 0.1 and NT = 0.08 were set as the default values. We parsed the xml files and then calculated the total numbers of actual vehicles in the data set. The number of vehicles in the statistical data set while using the YOLO-UA model was called the number from statistics. We calculated the accuracy, precision, and recall rate of the vehicle statistics and compared these with the results from the YOLO model. The experimental results are shown in Tables 1 and 2.

**Table 1.** YOLO general model traffic statistics.

| Data Sets | Number of Actual | Number from Statistics | Accuracy Rate (%) | Precision (%) | Recall (%) |
|---|---|---|---|---|---|
| Training verification set | 139,098 | 91,721 | 65.94 | 98.10 | 64.95 |
| Training set | 111,351 | 73,531 | 66.04 | 98.03 | 64.93 |
| Verification set | 27,747 | 18,190 | 65.56 | 98.05 | 64.47 |
| Test set | 34,760 | 22,936 | 65.98 | 97.92 | 64.81 |

**Table 2.** YOLO-UA general model traffic statistics.

| Data Sets | Number of Actual | Number from Statistics | Accuracy Rate (%) | Precision (%) | Recall (%) |
|---|---|---|---|---|---|
| Training verification set | 139,098 | 109,191 | 78.50 | 97.29 | 76.44 |
| Training set | 111,351 | 87,458 | 78.54 | 97.30 | 76.49 |
| Verification set | 27,747 | 21,727 | 78.30 | 97.27 | 76.24 |
| Test set | 34,760 | 27,743 | 78.95 | 96.11 | 76.76 |

From the results in Tables 1 and 2, we found that, under the presupposition that the actual vehicle base of the data set was large, when the statistical accuracy rate of the YOLO model network was compared with that of the YOLO-UA model network, it increased by 13%, the recall rate increased by 12%, and the recall rate increased at the expense of precision. When we evaluate the proposed model, a reduction in the number of missed checks could improve the recall rate, but a decrease in the number of missed inspections would increase the number of false detections, which results in a lower precision rate.

*4.4. Scene and Weather Adaptation Experiments*

After we adjusted the threshold parameters to test a series of different pictures, we found that, when CT and NT were 0.1 and 0.15, the detection results were optimal. To find how adaptable the optimized models were to traffic conditions in multiple scenes and various weather conditions, we made different data sets of over 1000 pictures of different scenes: sunny days, cloudy days, rainy days, and night, respectively. We used the YOLO and YOLO-UA models for testing, and the results are compared in Tables 3 and 4. Apparently, the accuracy rates in Table 4 are higher than those

in Table 2. The reason is that, in the training stage of deep neural network, GIOU represents the overlapping relationship between the network prediction frame and the real target frame in the figure, and the position deviation between the prediction frame and the real frame can be well reflected. It is based on the design of GIOU, as the network training time is increased, the location gap between the prediction frame and the real frame is decreased. Although the overlap region is increased, the total loss function of the network is decreased, and the accuracy and performance of the network location can be improved.

**Table 3.** YOLO multi-environment traffic statistics.

| Data Sets | Number of Actual | Number from Statistics | Accuracy Rate (%) | Precision (%) | Recall (%) |
|---|---|---|---|---|---|
| Sunny | 6823 | 4590 | 67.27 | 97.62 | 65.87 |
| Cloudy | 5922 | 4853 | 81.95 | 98.29 | 80.50 |
| Rainy | 8517 | 5461 | 64.11 | 99.44 | 63.05 |
| Night | 5731 | 4259 | 74.31 | 99.02 | 63.64 |

**Table 4.** YOLO-UA multi-environment traffic statistics.

| Data Sets | Number of Actual | Number from Statistics | Accuracy rate (%) | Precision (%) | Recall (%) |
|---|---|---|---|---|---|
| Sunny | 6823 | 6480 | 94.97 | 97.27 | 92.37 |
| Cloudy | 5922 | 5895 | 99.54 | 94.66 | 94.23 |
| Rainy | 8517 | 7106 | 83.53 | 96.36 | 80.40 |
| Night | 5731 | 5727 | 99.90 | 90.05 | 90.00 |

The average accuracy rate and the average recall rate of the YOLO-UA model were 22% and 21% greater when compared with those of the YOLO model, respectively, even under different detection conditions. These results prove that when the algorithm of the YOLO-UA model is substituted for that of the YOLO model, network performance is greatly improved. These results prove that the YOLO-UA model has good adaptability to different scene and weather conditions, and it can reliably complete the tasks that are required for traffic flow monitoring. However, there are more vehicles on the road during sunny day due to the influences of time and location, and the camera is farther away from them to capture more targets, which leads to the problems of smaller vehicle targets, unclear vehicles, and smaller workshop distance in the images. This problem makes the vehicle difficult to be detected by the model, and more vehicles are missed in the detection process. At night, the road vehicles are sparse, and the vehicle targets are large in the images, for that it is easy to be detected and counted by the model. After testing and observing pictures that were taken in various scenarios and weather conditions, as shown in Figure 5, we found the YOLO model to have serious issues with missed detections and false detections, which affected the statistical results.

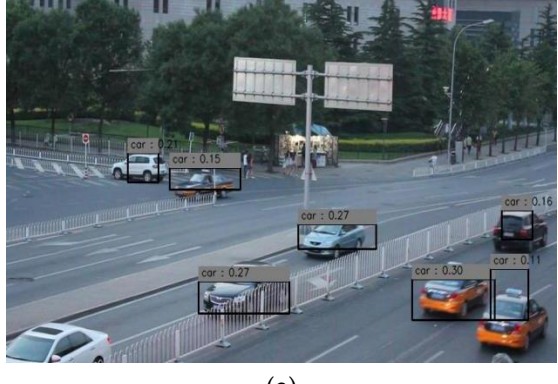 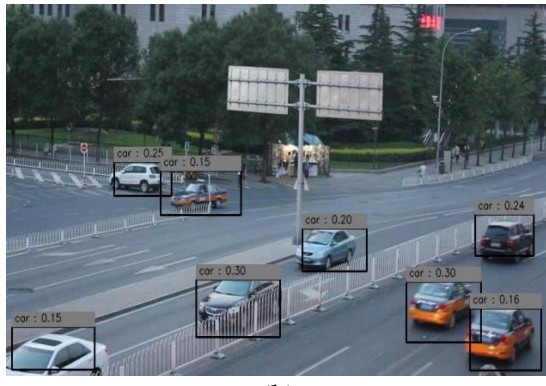

(**a**)  (**b**)

**Figure 5.** *Cont.*

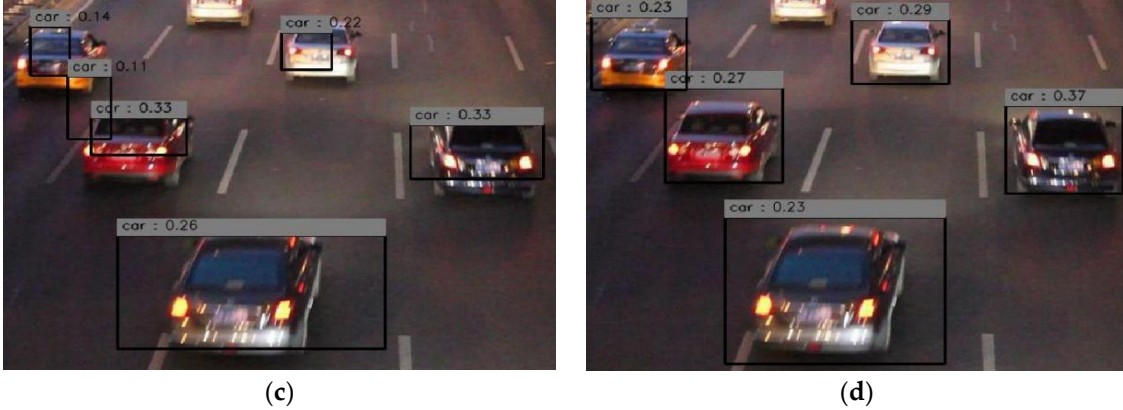

(**c**)　　　　　　　　　　(**d**)

**Figure 5.** Multi-scenario vehicle monitoring (**a**) YOLO cloudy day vehicle monitoring, (**b**) YOLO-UA cloudy day vehicle monitoring, (**c**) YOLO night vehicle monitoring, and (**d**) YOLO-UA night vehicle monitoring.

The YOLO model cannot predict multi-scale anchor boxes. Each grid only predicts two bounding boxes, which limits the target positioning with different sizes. This can easily cause missed statistics in vehicle monitoring, indirectly affecting the accuracy of the results. Optimization ameliorated the false detections and missed detections, and the resulting network generalizations were better.

### 4.5. Impact Accuracy Experiments

The CT value was changed from 0.08 to 0.30 and the NT value was changed from 0.08 to 0.20 to determine the effects of impact factors on statistical accuracy. Figure 6 shows the results, where the accuracy rates of the YOLO and YOLO-UA models for different data sets are shown as dashed and solid lines, respectively. The accuracy rates of the YOLO-UA model for different data sets are obviously higher than those of the YOLO model. With the increase in NT value, the statistical accuracy rate of the YOLO-UA model rose more rapidly than that of the YOLO model. When multiple vehicles are close to each other, the boundary frames of detected vehicles overlap. However, when we increased the NT value, the leakage detection problems did not occur when the Non-maximum suppression algorithm was run. We found that, when the CT value was increased and the category confidence was used to make classification judgments, it was easy to separate the correct frames from the less reliable ones that caused leakage statistics and led to a decrease in the statistical accuracy rate.

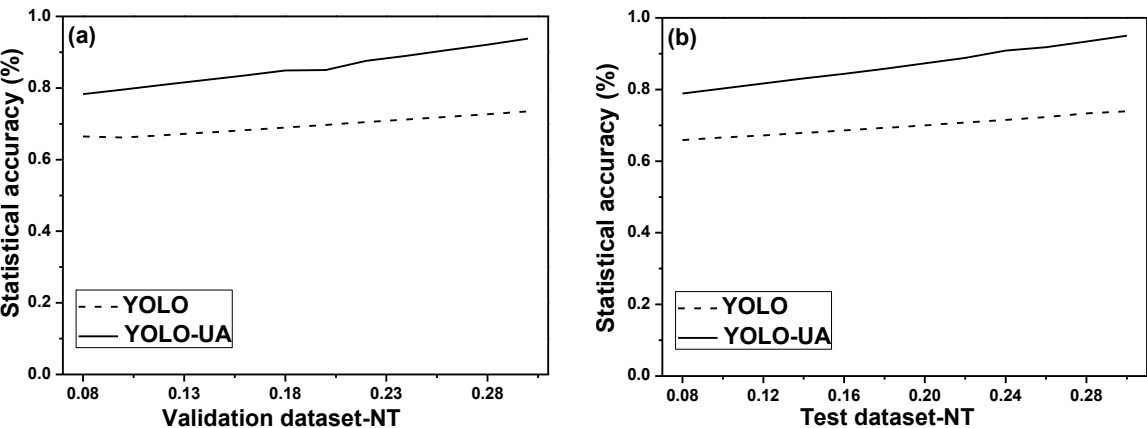

**Figure 6.** *Cont.*

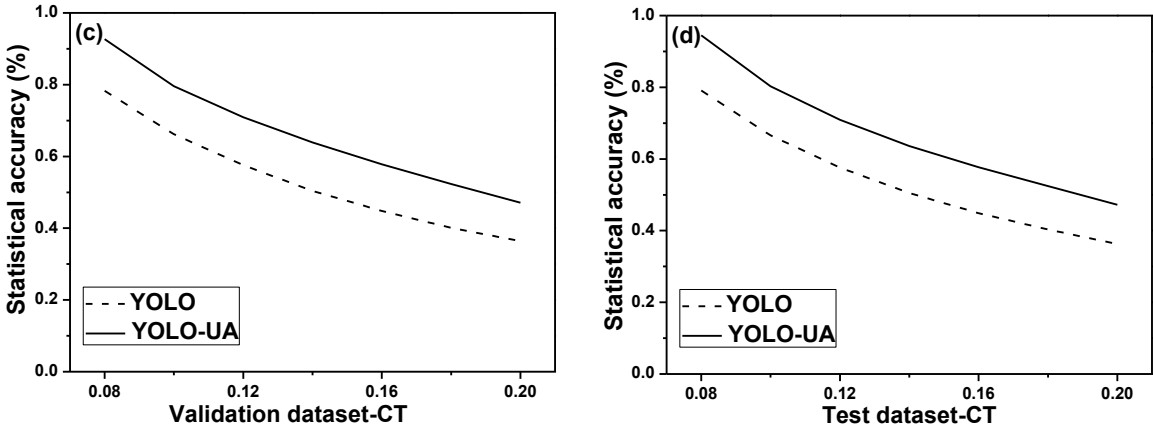

**Figure 6.** Effects of CT and NT impact factors on statistical accuracy, (**a**) NT on the accuracy of verification sets, (**b**) NT on the accuracy of test sets, (**c**) CT on the accuracy of verification sets, and (**d**) CT on the accuracy of test sets.

### 4.6. Online Video Traffic Test Experiment

We set NT at 0.08, adjusted the CT value to find the optimal parameters, and tested traffic videos that were taken using cell phones in different scenes and weathers to verify the effect of the YOLO-UA model video traffic flow monitoring. Each video was 35 s long. Video A was shot from the roadside on a sunny evening, video B was shot from an overpass at noon on a sunny day, and video C was shot from a sidewalk on a cloudy morning. Better test results were obtained at CT = 0.78, as the results in Tables 5 and 6 show.

**Table 5.** YOLO video traffic monitoring statistics.

| Video | Number of Actual | Number from Statistics | Accuracy (%) |
|---|---|---|---|
| A.mp4 | 20 | 16 | 80 |
| B.mp4 | 26 | 22 | 84.6 |
| C.mp4 | 13 | 11 | 84.6 |

**Table 6.** YOLO-UA video traffic monitoring statistics.

| Video | Number of Actual | Number from Statistics | Accuracy (%) |
|---|---|---|---|
| A.mp4 | 20 | 19 | 95 |
| B.mp4 | 26 | 26 | 100 |
| C.mp4 | 13 | 13 | 100 |

The testing results show that the YOLO-UA model took an average of 30 ms to count the vehicles in each frame. The CT value was set in the range of [0.76, 0.81], where the YOLO-UA traffic statistics were equal to the actual traffic numbers to determine the optimal parameter range and find the optimal CT value for multiple experiments. When we used the YOLO-UA model for traffic flow statistics, it was important that the camera positions not be too high and the field of view not be too large. If the positions were too high, they could cause missed detections. Incorrect detections occurred if the fields of view were too large. If these positions' problems were noted and adjusted, optimal statistical effects could be achieved by adjusting the threshold values. The statistical accuracy rates of traffic flow in Tables 5 and 6 are 84.6% and 100%, respectively, with an average statistical time of 30 ms. When compared with the single-Gaussian model and the ViBe algorithm [7], the YOLO-UA model had the shortest testing time and the highest accuracy rate, as the results in Table 7 show.

**Table 7.** Different algorithms used for traffic monitoring.

| Algorithm | Accuracy Rate (%) | Time (ms) |
|---|---|---|
| Single-Gaussian [6] | 93.5 | 118 |
| ViBe [7] | 96.2 | 158 |
| Faster R-CNN [21] | 90.8 | 85 |
| YOLO [31] | 84.6 | 30 |
| YOLO-UA | 100 | 30 |

## 5. Conclusions

Because of the limitations of the YOLO model, when multiple vehicles in a video are close to each other or occlude one another, positioning problems arise, which cause erroneous or leakage statistics that negatively affect the statistical accuracy. However, the YOLO algorithm that was optimized by GIOU yields highly accurate traffic flow statistics, and adjusting the threshold parameter achieved results that were extremely close to the actual number of vehicles. This optimized algorithm can reliably execute traffic flow monitoring and statistical analysis for multiple scenarios and in a variety of weather conditions, and it has a certain reference value. As the experimental results show, the YOLO-UA model is not ideal for rainy-day traffic monitoring and it has a low recall rate. Therefore, we will introduce (i) a deep residual network to replace YOLO's own network structure, (ii) focal loss to resolve category imbalances, and (iii) batch normalization during training to further improve its adaptability to various weather conditions.

**Author Contributions:** Investigation, C.-Y.C., J.-C.Z. and J.L.; Methodology, C.-Y.C. and J.-C.Z.; Formal analysis, C.-Y.C., J.-C.Z. and Y.-Q.H.; Writing—Original draft preparation, C.-Y.C., C.-F.Y., J.L. and J.-C.Z.; Writing—Review and editing, C.-F.Y., C.-Y.C. and J.-C.Z.

**Funding:** This work was supported by projects under No. MOST 108-2221-E-390-005 and MOST 108-2622-E-390-002-CC3.

**Conflicts of Interest:** The authors declare no conflict of interest.

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
