# Peer review of "Investigation of a Promoted You Only Look Once Algorithm and Its Application in Traffic Flow Monitoring"

_applsci, doi:10.3390/app9173619_

Round 1

Reviewer 1 Report

General Comments:

The paper presents an algorithm for vehicle detection using video feed. The proposed algorithm YOLO-UA is developed by improving the YOLO algorithm. The topic of the paper is interesting and in general, the paper is nicely motivated. However, paper needs to improve by proving more insights about the numerical result outcomes. In particular, I have following specific comments that needs to be addressed.

Specific comments:

P-7, L-258: "The precision and recall rates showed opposite trends." This statement needs explanation. P-7, L-279: "A After..." Here A is redundant. Please delete "A". P-8, Table-3: I am surprised to see night time accuracy higher than sunny time. It needs some discussion to justify. P-8, Table-4: The accuracy reported here is significantly higher than Table 2. It needs some discussion/justification. P-8, L-293-294: "These results prove that the YOLO model has good adaptability to different scene and weather conditions..." Whether Authors meant YOLO-UA model instead of YOLO model here? It will be helpful to provide the algorithm as appendix, or a GitHub link can be provided, it will be helpful if a researcher wants to test the algorithm.

Author Response

Reviewer 1

The paper presents an algorithm for vehicle detection using video feed. The proposed algorithm YOLO-UA is developed by improving the YOLO algorithm. The topic of the paper is interesting and in general, the paper is nicely motivated. However, paper needs to improve by proving more insights about the numerical result outcomes. In particular, I have following specific comments that needs to be addressed. Specific comments:

P-7, L-258: “The precision and recall rates showed opposite trends.” This statement needs explanation.

Answer: The increase in the statistics of correct vehicle detection could improve the recall rate, but as vehicle misdetection rate increased, the detection’s precision would be decreased. On the contrary, the reduction in the number of false vehicle detection was conducive to improve the detection’s precision. However, the algorithm could reduce the number of false vehicle detection only when strict judgment conditions were set, but at the same time, it also led to a reduction in the correct statistics of the vehicle, resulting in a lower recall rate. For that, the precision and recall rates showed opposite trends. Please see lines 291~297.

P-7, L-279: "A After..." Here A is redundant. Please delete "A".

Answer: Thanks for reviewer's carefulness, “A” is deleted, please see line 318.

P-8, Table-3: I am surprised to see night time accuracy higher than sunny time. It needs some discussion to justify.

Answer: However, due to the influences of time and location, there are more vehicles on the road during sunny day, and the camera is farther away from them to capture more targets, which leads to the problems of smaller vehicle targets, unclear vehicles, and smaller workshop distance in the images. This problem makes the vehicle difficult to be detected by the model, and more vehicles are missed in the detection process. At night, the road vehicles are sparse, and the vehicle targets are large in the images, for that it is easy to be detected and counted by the model. Please see lines 341~347.

P-8, Table-4: The accuracy reported here is significantly higher than Table 2. It needs some discussion/justification.

Answer: The reason is that in the training stage of deep neural network, GIOU represents the overlapping relationship between the network prediction frame and the real target frame in the images, and the position deviation between the prediction frame and the real frame can be well reflected. It is based on the design of GIOU, as the network training time is increased, the location gap between the prediction frame and the real frame is decreased. Although the overlap region is increased, the total loss function of the network is decreased, and the accuracy and performance of the network location cab be improved. Please see lines 323~330.

P-8, L-293-294: "These results prove that the YOLO model has good adaptability to different scene and weather conditions..." Whether Authors meant YOLO-UA model instead of YOLO model here?

Answer: Thanks for reviewer’s carefulness, it’s really YOLO-UA model instead of YOLO model. I have made changes, please see line 340.

It will be helpful to provide the algorithm as appendix, or a GitHub link can be provided, it will be helpful if a researcher wants to test the algorithm

Answer: I am very sorry, I also want to open the algorithm in order to facilitate the research of the researchers, but this is a team project with the company, I am not allowed and have no right to disclose the algorithm code. Sorry again.

Reviewer 2 Report

This paper proposed an approach for real-time monitoring of traffic-flow problem.  Authors used the original YOLO network and made fine tuning for optimization (for car detection). 

In network model aspect, there was almost no contribution. Authors considered IOU to the loss function.  From Eqs. (1)~(4), they evaluated GIOU as a parameter for loss function. I think all equations should be numbered and they referenced in the sentences.  In Eq. (5), what is 'w'? I think we need more detailed explanation. Authors mentioned "GIOU has the merit of not being sensitive to scale,".  But there is no exact description and evidence in the manuscript.  I think more exact evaluation is needed from Eq. (5) to (9). Some part was not unclear. In Fig. 6, how authors selected CT and NT exactly? I could not understand how to select from Fig. 6. For example, CT=0.78 was not in the range of Fig. 6.  How did authors get this? In Table 7, authors would be better to give the reference number to each existing method. Major problem is that what is main contribution. Authors should address their contribution exactly. It would be better to add some related works: - Receptive Field Block Net for Accurate and Fast Object Detection,  • Songtao Liu • Di Huang • Yunhong Wang, ECCV 2018. - Pelee: A Real-Time Object Detection System on Mobile Devices, • Jun Wang • Tanner Bohn • Charles Ling, NeurIPS 2018  -DeNet: Scalable Real-time Object Detection with Directed Sparse Sampling, • Lachlan Tychsen-Smith • Lars Petersson, ICCV 2017  - Fast Multi-feature Pedestrian Detection Algorithm Based on Discrete Wavelet Transform for Interactive Driver Assistance SystemMultimedia Tools and Applications, Vol.75, pp.15229–15245, 2016.

Author Response

This paper proposed an approach for real-time monitoring of traffic-flow problem. Authors used the original YOLO network and made fine tuning for optimization (for car detection). In network model aspect, there was almost no contribution. Authors considered IOU to the loss function.  From Eqs. (1)~(4), they evaluated GIOU as a parameter for loss function. I think all equations should be numbered and they referenced in the sentences. 

Answer: Thanks for your comments. We have made changes to each equation and numbered it.

In Eq. (5), what is 'w'? I think we need more detailed explanation.

Answer: I am sorry to bring you doubts, due to spelling mistakes should be . is the weight value of neurons in the neural network. These weight values can be used to predict the category and position coordinates of the target, please see equation (11).

Authors mentioned "GIOU has the merit of not being sensitive to scale,".  But there is no exact description and evidence in the manuscript.  I think more exact evaluation is needed from Eq. (5) to (9).

Answer: Thank you for pointing out that I made the following modifications:

GIOU can be regarded as a parameter of overlap measurement, which reflects the overlap relationship between prediction box and real box, please see lines 215~216.

Some part was not unclear. In Fig. 6, how authors selected CT and NT exactly? I could not understand how to select from Fig. 6. For example, CT=0.78 was not in the range of Fig. 6. How did authors get this?

Answer: Figure 6 explores the impacts of CT and NT change models on the detection statistics of image data, which was obtained remotely using cameras. Here, the video obtained by the close-range mobile phone is detected and counted. Because of the large target of the short-range shooting, only when the strict judgment condition is set, there will be fewer false detections on vehicles. The CT value can be continuously adjusted and tested the detected statistical vehicles. The experimental results show that when CT is between 0.76 and 0.8, there are fewer false vehicle detection and the model statistics are closest to the real number of vehicles.

In Table 7, authors would be better to give the reference number to each existing method.

Answer: I have given a reference number. The experiment was reworked on the platform and the conclusion was reached, please see table 7.

Major problem is that what is main contribution. Authors should address their contribution exactly.

Answer: In order to solve the problem of poor positioning of the YOLO model and low accuracy of vehicle statistics, the method of fine-tuning the model structure and the GIOU optimization loss function is proposed to enhance the accuracy of the target positioning. After optimization, it can be more reliably applied to video vehicle statistics in real-time and actual scenes. Please see lines 106~110.

It would be better to add some related works: - 

Receptive Field Block Net for Accurate and Fast Object Detection,  • Songtao Liu • Di Huang • Yunhong Wang, ECCV 2018.

 Pelee: A Real-Time Object Detection System on Mobile Devices, • Jun Wang • Tanner Bohn • Charles Ling, NeurIPS 2018 

-DeNet: Scalable Real-time Object Detection with Directed Sparse Sampling, • Lachlan Tychsen-Smith • Lars Petersson, ICCV 2017 

- Fast Multi-feature Pedestrian Detection Algorithm Based on Discrete Wavelet Transform for Interactive Driver Assistance System, Multimedia Tools and Applications, Vol.75, pp.15229–15245, 2016.

Answer: Thanks for your suggestion. I have already adopted the proposal and added some related work. Please see lines 81~100 and references 22~27.

Round 2

Reviewer 2 Report

This paper has been well changed based on the comments. So I recommend this to be accepted with its current form.